# Left ventricular segmentation method based on optimized UNet and improved CBAM: ESV and EDV tracking study

Kerang Cao[1,2], Miao Zhao[1,2], Minghui Geng[1,2], Shuai Zheng[1,2], Hoekyung Jung[3]*

**1** College of Computer Science and Technology, Shenyang University of Chemical Technology, Shenyang, China, **2** Key Laboratory of Intelligent Technology of Chemical Process Industry in Liaoning Province, Shenyang, China, **3** Computer Engineering Department, Paichai University, Daejeon, Korea

* hkjung@pcu.ac.kr

## Abstract

This paper introduces an optimized nested UNet model for automated left ventricular segmentation in cardiac function assessment. We utilize the EchoNet-Dynamic dataset, which contains both video data and expert annotations. Unlike conventional methods such as DeepLabv3 that struggle with large model sizes and imprecise segmentation, Our proposed model introduces a deeper feature extraction module to effectively capture multi-scale features and reduce computational overhead. By integrating the CBAM (Attention module) attention mechanism and a lightweight SimAM (Simple Attention Module) module, we enhance feature selectivity and minimize redundancy. To further stabilize training and address gradient issues, we combine binary cross-entropy and Dice loss functions. Experimental results reveal that our model significantly outperforms existing methods, achieving a 1.05% increase in the Dice coefficient and reducing model size to 15% of the original. These improvements not only enhance the accuracy of cardiac function assessments but also provide a more efficient solution for automated diagnosis in clinical practice.

## Introduction

According to the World Health Organization, heart disease remains one of the leading causes of death in humans [1] Accurate and effective early diagnosis is therefore crucial for improving heart disease treatment outcomes and reducing mortality rates. In clinical settings, clinicians assess ventricular function by obtaining rapid and precise cardiac images. These images support the calculation of essential clinical indicators, including ventricular volume, ejection fraction (EF), and myocardial mass [2].

Echocardiography has become a widely used method for acquiring cardiac information due to its ability to generate spatiotemporal data in the form of videos. This approach enables clinicians to visualize dynamic changes within the heart and measure functional parameters such as EF, which reflect hemodynamic status and

**Data availability statement:** The dataset used is a public dataset from Stanford University, available at: https://stanfordaimi.azurewebsites.net/datasets/834e1cd1-92f7-4268-9daa-d359198b310a According to the usage agreement for the EchoNet-Dynamic dataset: "The EchoNet-Dynamic dataset may be freely viewed and used, but the download link must not be shared with others. If other users within your organization wish to use the EchoNet-Dynamic dataset, they must register as individual users and comply with all terms of this research use agreement." We recommend registering for an account through the official EchoNet-Dynamic website to access the publicly available content and dataset details. The original data is also provided as a Supporting Information file.

**Funding:** This work was supported by the Institute of Information & Communications Technology Planning & valuation(IITP)-Innovative Human Resource Development for Local Intellectualization program grant funded by the Korea government(MSIT)(IITP-2025-RS-2022-00156334,contribution rate:70%);Basic scientific research general project of Liaoning Provincial Department of Education in 2023(JYTMS20231518)

**Competing interests:** The authors have declared that no competing interests exist.

myocardial contractility. Notably, a lower EF often indicates more severe ventricular remodeling and diminished myocardial contractile function. However, poor-quality echocardiographic images can result in inaccurate EF measurements, delaying appropriate treatment and negatively affecting patient outcomes. Given these challenges, there is a growing need for automated and accurate assessment of left ventricular ejection fraction. Traditional echocardiographic analysis demands considerable time and expertise from cardiologists. In response, developers have introduced automated cardiac analysis algorithms to improve efficiency and accuracy. Early segmentation methods for the left ventricle focused on mathematical models, which achieved only limited success and were often validated solely on private datasets [3].

With the emergence of deep learning, however, echocardiographic analysis has advanced significantly, especially as these new methods can be trained and evaluated on larger public datasets. The emergence of deep learning has significantly advanced the field of echocardiographic analysis. The successful application of convolutional neural networks (CNNs) on large datasets such as ImageNet and ADE20K has led to deep learning methods becoming the mainstream solution for left ventricular segmentation in echocardiography. Furthermore, the public release of large-scale echocardiographic datasets like EchoNet-Dynamic [4] has enabled more effective training and evaluation of these models.

Recently, Vision Transformer (ViT) models [5] have demonstrated strong potential in computer vision and echocardiographic segmentation tasks. For example, Deng et al. [6] introduced a TransBridge network—composed of two CNNs connected by Transformer blocks—that enhances the model's capacity to perceive multi-scale structures in ultrasound images through a bridging module. Additionally, Zeng et al. [7] proposed the MAEF-Net model, which incorporates multiple attention mechanisms to help the network focus on key image regions, especially those associated with the left ventricle, thus improving segmentation accuracy.

Building on these advances, Liao M et al. [8] explored the strengths of Transformer-based models for segmentation. They proposed two different architectures: one combining Swin Transformer with K-Net, and another using Segformer. Evaluations on the EchoNet-Dynamic dataset showed that these models achieved a Dice coefficient of 92.92%, even on challenging samples.

Despite these advances, Transformer-based models often require substantial computational resources, have slower inference speeds, and demand high-end hardware, which can restrict their use in some institutions and enterprises. Consequently, researchers now face the challenge of improving left ventricular segmentation accuracy while simultaneously reducing model complexity and hardware requirements to enable broader practical adoption.

### Contribution to research

To solve this problem, this paper proposes an improved nested UNet model. The model incorporates a deeper feature extraction module (NestedUNet block) to enhance feature representation and enable multi-scale feature capture. At the same time, the UNet network is tailored to achieve a lightweight design, thereby reducing

computational and storage overhead. Additionally, to reduce parameter redundancy, improve computational efficiency, and enhance local information, this paper incorporates the lightweight attention mechanism SimAM [9] and integrates it with CBAM. We evaluate our model on the EchoNet-Dynamic dataset for segmenting left ventricular structures during end-diastolic and end-systolic phases. The model achieves a Dice coefficient of 93.16% while maintaining a compact parameter size.

Novelties of this study include:

- An improved nested UNet structure is proposed to enhance feature expression and capture multi-scale features by introducing a deeper feature extraction module (NestedUNet block).

- The UNet network has been trimmed and designed to be lightweight to reduce computational and storage overhead.

- CBAM is fused with SimAM, a lightweight attention mechanism, to reduce parameter redundancy, improve computational efficiency, and enhance local information.

- Binary cross-entropy loss and Dice loss are combined to solve the problem of vanishing gradients or gradient explosions.

These innovations strike a balance between segmentation accuracy and computational efficiency, offering a robust solution for echocardiographic analysis.

## Related research

In this section, some of the related work will be discussed. First, a brief review of previous ventricular segmentation methods, with a focus on CNN-based methods, is provided. Second, the recently popular Vision Transformer (ViT) and the basic deep learning networks related to the research in this paper are introduced.

### Non-in-depth learning methods

Early non-deep learning models of left ventricular segmentation primarily targeted the identification and delineation of left ventricular membrane boundaries. Methods such as active contours [10] have achieved relatively effective segmentation in ultrasound images, but they rely on specific data formats and lacked scalable. Barbosa et al. [11] proposed a global anatomical affine optical flow and local recursive block matching technique. Their method used affine optical flow to capture the overall ventricular motion and employed recursive block matching to refine feature tracking in complex scenarios. Bernard et al. [12] compared nine segmentation methods on a relatively fair basis by evaluating the same dataset (RT3DE for 45 videos), including multiple metrics such as segmentation precision, computational time, and robustness. The experiments show satisfactory results, which proves the competitiveness of the method of Barbosa et al. Nonetheless, these traditional models still produce results that often deviate from cardiologist assessments, and they struggle to maintain robustness across diverse segmentation tasks on larger datasets.

### In-depth learning methods

The widespread application of deep learning technology has driven the latest advances in the field of medical imaging. Convolutional neural network-based cardiac segmentation [13–15], three-dimensional convolutional network (3D CNN) [16], multi-fusion network [17,18], and variants of residual networks have become the most commonly used methods for various ventricular segmentation tasks.

For example, Baumgartner et al. [19] discussed the application of 2D and 3D deep learning techniques in cardiac magnetic resonance imaging (MRI) image segmentation and compared the effects of the two methods when processing cardiac MRI images. Their research clearly highlights the differences between network architectures in capturing local

features and global contextual information, offering guidance for future UNet-related studies. Patravali et al. [20] proposed a cardiac MRI segmentation method combining 2D and 3D fully convolutional neural networks (FCNs), which uses 2D networks to capture local features and volumetric context information through 3D networks to improve segmentation accuracy. These studies reveal the important influence of network architecture on segmentation results.

In recent years, various deep learning network architectures have shown distinct strengths in medical image segmentation tasks. Among these, the fully convolutional neural network M-Net architecture proposed by Jang et al. [21] has demonstrated effective segmentation of the left ventricle, right ventricle, and myocardium, highlighting its considerable potential for clinical application. This finding contrasts with the study by Luo et al. [22], which used the UNet architecture to improve the segmentation of cardiac MRI by introducing a context extraction module and a segmentation module, combined with the information of the previous segmentation labels. These improvements ensure effective feature learning through the design of skip connections, reducing feature loss and gradient dispersion. Huang et al. [23] further optimized the segmentation effect by using full-scale skip connections to combine low-level details with high-level semantics of feature maps at different scales on the basis of UNet. The Double-UNet proposed by Jha et al. further improved the performance on various medical image segmentation datasets by stacking two UNets [24]. Moreover, to address the challenges posed by the computational complexity of the model, Liao et al. [25] proposed LightM-UNet, which is more suitable for deployment in resource-constrained environments by introducing a lightweight design that reduces computational and storage overhead while maintaining high-precision segmentation performance.

These findings collectively indicate that, despite the success of a large number of works based on other architectures, UNet and its variants are still evolving and showing great potential for medical image segmentation, especially in response to parameter changes and optimizing the use of computing resources.

In addition to CNN-based approaches, recent advances in the field of computer vision have also leveraged Transformer architectures. The recent success of Vision Transformer (ViT) [26] has led to the development of deep learning-based approaches in the field of computer vision, mainly by introducing attention mechanisms to handle a variety of tasks. After extensive pre-training, ViT achieves classification accuracy on the ImageNet dataset that rivals ResNet. However, as the complexity of these models increases, so does the need for their computing resources.

To address this challenge, Liu et al. [27] in 2021 proposed a hierarchical Transformer that uses a shifted window to compute. With this shift window scheme, the calculations are confined to non-overlapping local windows while allowing connections across windows, thus increasing efficiency while maintaining high accuracy. Cao et al. [28] implemented the Swin Transformer module in a UNet-like architecture and developed a multi-organ medical image segmentation framework for MR images, validating its effectiveness in segmentation tasks.

Although these emerging Transformer models excel in accuracy, their application in clinical settings faces challenges. Hatamizadeh et al. [29] noted that they often have high computational complexity and resource requirements. These characteristics result in the need for more computing resources, such as CPUs or GPUs, during the training and inference process of the model, which not only increases the cost of deployment, but also poses a challenge to real-time diagnosis and treatment in the hospital environment. Especially in clinical settings, timely feedback is essential for rapid diagnosis and treatment; however, the long inference time of the Transformer model may affect the efficiency of the physician and the treatment time of the patient. Therefore, although the Transformer model shows excellent performance in the research stage, its high computational complexity and inference time may lead to a delay in the system response and affect the decision-making process of doctors in practical applications, especially in real-time image analysis or auxiliary diagnosis systems. As a result, even if Transformer models perform well in theory, their feasibility in practical applications still faces significant challenges.

In summary, there is a clear need to balance segmentation accuracy with computational efficiency for clinical deployment. Therefore, based on the above questions, this study was inspired by [23–25] and the improved UNet was considered as a model architecture for the left ventricular segmentation task in echocardiograph. The aim is to address the

computational complexity and inference time requirements in practical applications—especially in real-time image analysis or auxiliary diagnosis systems—thus improving the work efficiency of doctors and reducing the treatment time for patients.

### Heart segmentation algorithm

This section presents a theoretical overview of the proposed encoder-decoder-based architecture. A comprehensive architectural design is provided, and the main building blocks and their functions are described.

### An overview of the model structure

In this study, we construct a fusion model combining an improved UNet architecture (Network for Image Segmentation) and the SCBAM attention module for segmentation of left ventricular end-diastolic volume (EDV) and end-systolic volume (ESV) in echocardiography using a video dataset. The overall structure and workflow illustrates in Fig 1, providing a clear overview of the main network components. Fig 1 shows the overall network structure, which mainly includes DA-UNet++, NestedUNet, VGG module, and SCBAM. Each of these modules plays a specific role in the network.

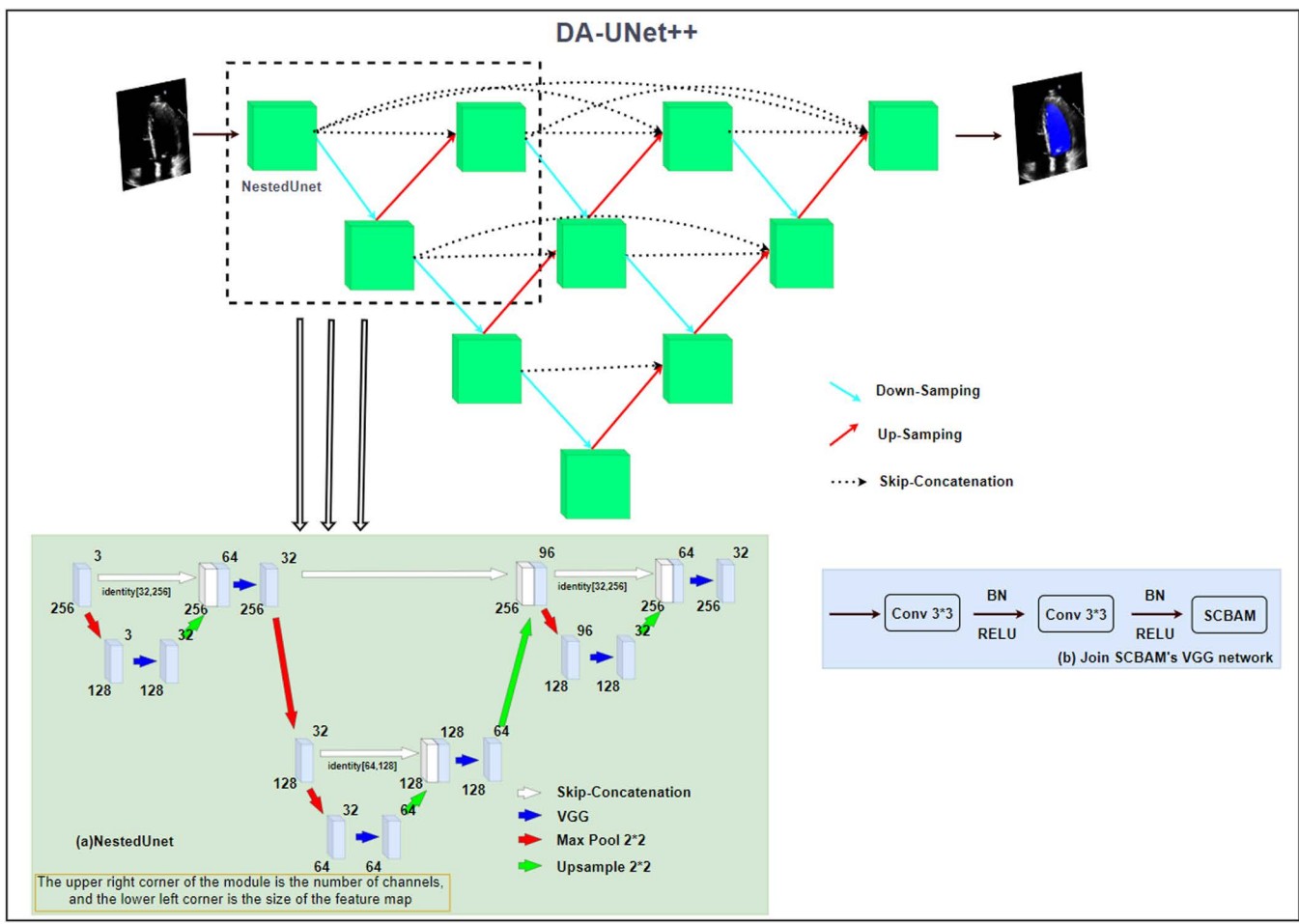

**Fig 1. Network structure model.** The network structure combines NestedUNet, the VGG module, and SCBAM. Its core is DA-UNet++, which is based on an encoder-decoder structure.

The network is based on a codec structure that processes images in DA-UNet++ by encoding (compressing information) and decoding (recovering information) features layer by layer. NestedUNet further enhances this encoding and decoding capability. The nested structure processes the feature maps of different resolutions through pooling operations and upsampling operations, ensuring that features of different scales ensure effective use, and passing the features in the encoder directly to the decoder through skip connections, helping to retain the high-resolution details of the nested structure. Thus, the network achieves efficient multi-scale feature extraction and high-resolution reconstruction. This is essentially the encoding and decoding process of UNet; the encoder extracts features layer by layer and the decoder recovers the spatial resolution layer by layer.

Feature extraction is further enhanced through the integration of the VGG (Deep Convolutional Neural Network Model) module. The VGG(Deep Convolutional Neural Network Model) module is used for feature extraction by stacking convolutional layers and batch normalization layers, adding SCBAM to further enhance the feature representation, each convolution operation keeps the spatial dimension of the input image unchanged, the number of channels of the output feature map changes according to the network configuration, the convolution operation can extract the edge information and feature information of the image, and the batch normalization helps to stabilize the training process.

The SCBAM module introduces attention mechanisms to focus on the most informative features. The SCBAM module combines channel attention, energy function, and spatial attention, with channel attention manipulation followed by SimAM [9]. The importance of each neuron is measured, and then spatial attention manipulation is performed to enhance feature representation in multiple dimensions.

## DA-UNet++ network structure

The DA-UNet++ model introduces optimizations in many aspects on the basis of UNet++ to further improve the performance of medical image segmentation tasks [30] and significantly reduce the computational and storage overhead. UNet++ improves the encoder and decoder structure of traditional UNet, introduces dense connections and multi-scale feature fusion, thereby improving the segmentation performance. Specifically, UNet++ employs dense hopping connections to transfer information not only between encoders and decoders, but also between feature maps of the same resolution. The multi-scale feature fusion strategy in the decoder combines features from different levels to capture image details and global information more accurately, thereby improving the accuracy of segmentation. To further optimize the model, researchers applied structural pruning to UNet++, especially after the introduction of deep supervision, each sub-network (L1~L4) [31] can output the segmentation results independently, which makes it possible to prune unnecessary parts in the testing stage, greatly reducing the network size, and reducing computing and storage overhead. The feature fusion operation in the first layer can be represented as:

$$Y_i^l = g([Y_0^l, Y_1^l, \ldots, Y_{i-1}^l, \tilde{X}_i^l]) \tag{1}$$

Thereinto, g indicates a feature fusion operation, $\tilde{X}_i^l$ Represents a feature map passed from the encoder.

On this basis, we propose an improved nested UNet structure (NestedUNet) is proposed. Through a nested design, it enhances segmentation performance, the network is able to capture more expressive features. First, $1 \times 1$ convolution is performed on the input to adjust the number of channels and preserve the original input information. Then, the spatial dimension of the input is halved by the maximum pooling layer, and the feature map feed into the first VGG module for two convolution operations. Then, the spatial dimension of the feature map was doubled by transposing the convolution for upsampling, and the upsampled feature map is concatenated with the previously saved map in the channel dimension, and the convolution operation was performed twice in the second VGG module again. At the same time, the SCBAM module is introduced for optimization. Nested UNet captures richer contextual information by fusing feature maps at different scales through multi-level pooling and upsampling operations. This

multi-scale feature fusion strategy significantly improves the model's performance in segmentation tasks, especially in object edge recognition and complex detail handling. Meanwhile, the Nested UNet architecture extracts hierarchical features through convolutional layers at different depths, capturing both local spatial information and preserving global spatial information. These features it extracts layer by layer, from low-level details (such as edges and textures) to high-level semantic information (such as objects and segmented regions), reflecting different levels of abstraction and further enhancing segmentation accuracy. In addition, the introduction of the VGG module enables the model to extract and retain more low-level edge information, reduce the ambiguous area in the segmentation result, and further improve the accuracy.

In summary, a more efficient DA-UNet++ model is proposed by optimizing the structure and lightweight design of UNet. By cropping redundant parts, introducing a deeper feature extraction module, strengthening multi-scale feature fusion, and adding VGG modules at key locations, the optimized model shows excellent performance in medical image segmentation tasks, which not only improves the segmentation accuracy, but also significantly reduces the computing and storage overhead, providing the possibility for deployment in the actual medical environment.

## Attention mechanism SCBAM

SCBAM combines the advantages of CBAM and SimAM to reduce parameter redundancy, improve computational efficiency, and enhance local information without increasing the model size, as illustrated in Fig 2. Specifically, the Channel Attention (CA) module enhances overall performance by highlighting important channel features in the feature map. The SimAM module further refines and smooths local features, while the Spatial Attention (SA) module strengthens the spatial information in the feature map.

CBAM enhances feature representation by extracting both channel and spatial attention features. The channel attention mechanism focuses on the most important channels, improving the representation of key feature channels. The spatial attention mechanism, on the other hand, targets specific regions of the image, thereby enhancing spatial features. The channel attention mechanism helps the model concentrate on the most relevant features for the task, while spatial attention emphasizes critical areas of the image.

SimAM, in contrast, emphasizes the local information of each pixel in the feature map. It highlights important features while reducing noise interference through a parameter-free energy function calculation. This localized focus eliminates the need for global computations, thereby improving computational efficiency. SimAM adjusts the attention of each feature map by calculating its variance, enabling the model to focus adaptively on areas with higher variance, typically associated with edges or transitions between different structures. These mechanisms help the model focus on more meaningful information at various levels, improving segmentation accuracy.

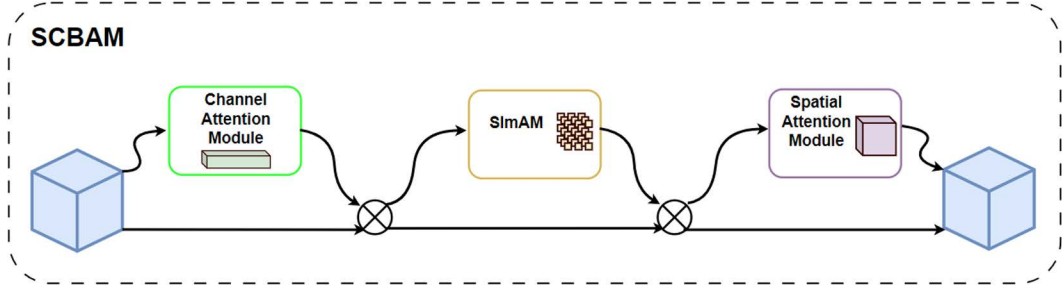

**Fig 2. SCBAM structure diagram.** SCBAM fuses the SimAM and CBAM attention modules to reduce parameter redundancy, improve computational efficiency, and enhance local information, all without increasing the model size.

By combining these attention mechanisms, the model can more effectively focus on key regions and important channels in the image, thereby improving segmentation accuracy. To prevent the SCBAM attention mechanism from significantly reducing the model's inference speed, this study employs the following optimization methods:

- Pruned Nested UNet Structure: By reducing the number of convolutional layers and pruning redundant convolutional kernels and neurons, the computational overhead is decreased, the model size is reduced, and inference speed is enhanced.

- Adjusting the Kernel Size of the Attention Mechanism: When using the spatial attention module of CBAM, smaller convolutional kernels are employed to reduce the computational burden while maintaining effective attention mechanisms.

## CBAM

The Convolutional Block Attention Module (CBAM) is an attention mechanism designed for convolutional neural networks. It combines channel attention and spatial attention to enhance the representational power of feature maps. CBAM improves the performance of the model by adaptively weighting the input feature map to highlight the important features and suppress irrelevant ones. CBAM consists of two sub-modules: the Channel Attention Module (CAM) and the Spatial Attention Module (SAM), as shown in Fig 3.

CBAM is similar to how our brains focus attention on important parts when observing something. Imagine you are looking at a busy street filled with people, vehicles, and buildings. If you are searching for a friend, your brain naturally concentrates on the crowd, ignoring irrelevant elements like parked cars or buildings. This is exactly what CBAM does in neural networks—it helps the model focus on "important" features, such as distinguishing people from vehicles in an image.

CAM captures the relationship between channels through global average pooling and global maximum pooling, generates channel attention, and applies it to the input feature map $\mathbf{F} \in \mathbb{R}^{C \times H \times W}$, The following steps are used to calculate the Channel Attention module:

Global average pooling and global maximum pooling

$$F_{\text{avg}}(c) = \frac{1}{H \times W} \sum_{i=1}^{H} \sum_{j=1}^{W} \mathbf{F}(c, i, j)$$

(2)

$$F_{\text{max}}(c) = \max_{i,j} \mathbf{F}(c, i, j)$$

(3)

Where H and W represent the height and width of the feature map.

Shared network: The two descriptors are passed through a shared multilayer perceptron (MLP) to generate a channel attention map $\mathbf{M}_c \in \mathbb{R}^{C \times 1 \times 1}$.

$$M_c = \sigma(\text{MLP}(\mathbf{F}_{\text{avg}}) + \text{MLP}(\mathbf{F}_{\text{max}}))$$

(4)

where $\sigma$ is the Sigmoid activation function, and MLP consists of two fully connected layers, and the output dimension of the first fully connected layer is $C/r$ (r is the rate of reduction), The output dimension of the second fully connected layer is $C$.

Attention Weighting: Multiply the input feature map by the channel attention map.

$$F'' = \mathbf{M}_s \odot F'$$

(5)

Thereinto, $\odot$ Represents multiplication by element.

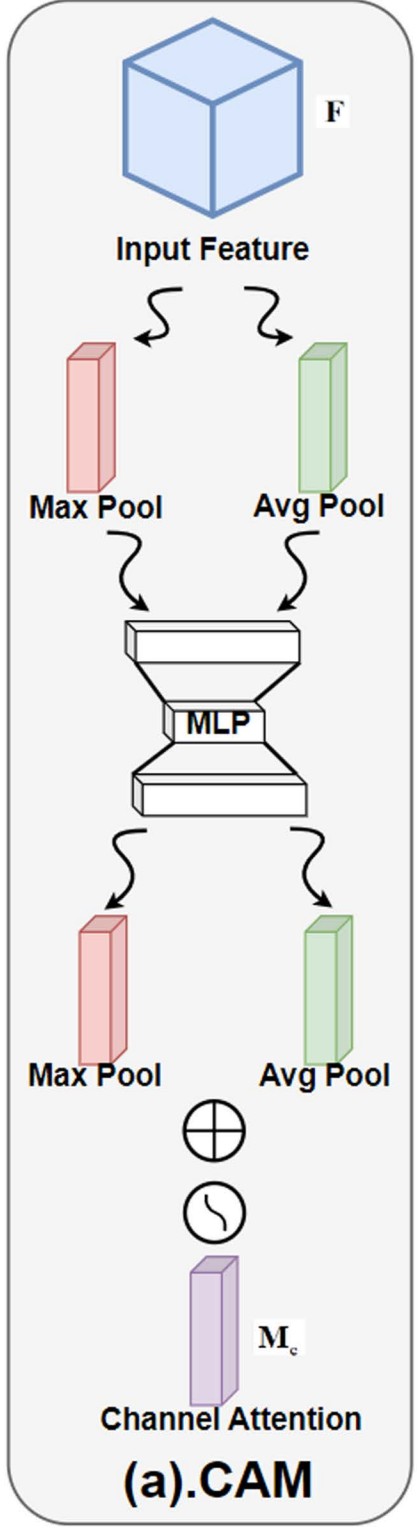

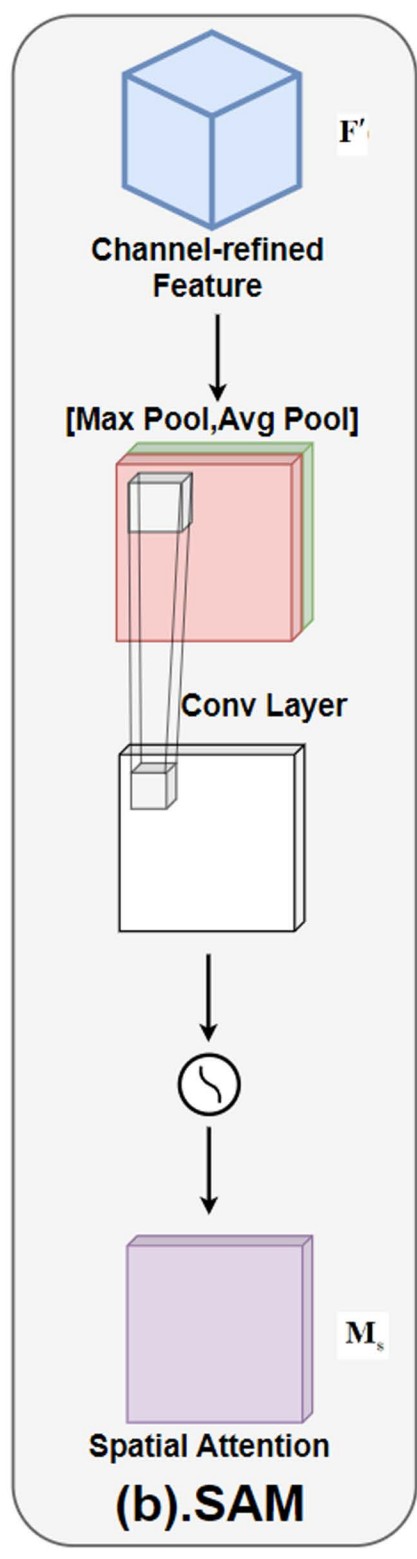

**Fig 3. CBAM structure diagram.** CBAM (Convolutional Block Attention Module) enhances the representation ability of feature maps by combining channel attention and spatial attention.

The SAM module captures the relationship between spatial locations through global average pooling and global maximum pooling along the channel dimension to generate a spatial attention map. Given the input feature map, the calculation steps of the Spatial Attention Module are as follows:

The global average pooling and global maximum pooling in the channel direction give two descriptors $\mathbf{F}'_{avg} \in \mathbb{R}^{1 \times H \times W}$ and $\mathbf{F}'_{max} \in \mathbb{R}^{1 \times H \times W}$.

$$F'_{avg}(i,j) = \frac{1}{C} \sum_{c=1}^{C} \mathbf{F}'(c,i,j)$$

(6)

$$F'_{max}(i,j) = \max_c \mathbf{F}'(c,i,j)$$

(7)

Join and convolutional operations: Connect these two descriptors in the channel dimension and generate a spatial attention map through the convolutional layer $\mathbf{M}_s \in \mathbb{R}^{1 \times H \times W}$.

$$M_s = \sigma(\text{Conv}([\mathbf{F}'_{avg}; \mathbf{F}'_{max}]))$$

(8)

Where $\sigma$ is the Sigmoid activation function, $[;]$ represents the connection operation at the channel dimension, Conv is a convolutional layer of $7 \times 7$.

Attention weighting: Multiply the input feature map by the spatial attention map.

$$F'' = \mathbf{M}_s \odot \mathbf{F}'$$

(9)

The overall architecture of CBAM consists of the two modules mentioned above, which generate the final weighted feature map by first performing channel attention and then spatial attention.

## SimAM

The SimAM (Simple, Parameter-Free Attention Module) [9] draws its design inspiration from neuroscience theory. It aims to enhance the representational capacity of convolutional neural networks while maintaining a lightweight structure without introducing additional parameters. SimAM addresses two major limitations of existing attention modules: first, their attention weights are typically restricted to either the channel or spatial dimension; second, they often require extra parameters, reducing flexibility and increasing model complexity.

Imagine being in a classroom where a teacher is explaining a topic. Some students (features) are more focused because they have studied more and understood the material more deeply. SimAM is like a teacher who can recognize which students are more attentive and concentrate more attention on those students, thereby enhancing the overall learning effectiveness. It gives more attention to students who have a deeper understanding (important features) while ignoring those who are less engaged (irrelevant features).

The SimAM module calculates attention weights by optimizing an energy function. The specific steps are as follows:

Definition of the energy function: Inspired by the concept of spatial inhibition in neuroscience, SimAM defines an energy function that quantifies the importance of each neuron.

$$e^*(t) = \frac{4(\sigma^2 + \lambda)}{(t - \mu)^2 + 2\sigma^2 + 2\lambda}$$

(10)

Thereinto, $\mu$ and $\sigma$ are the mean and variance of the channel, respectively, $\lambda$ is a hyperparameter.

Computation of attention weights: The importance weights of each neuron are calculated by solving the closed-form solution of the energy function.

$$E_{\text{inv}} = \frac{d}{4(v + \lambda)} + 0.5 \tag{11}$$

Thereinto, $d$ is the sum of squares of the deviations of the feature map, $v$ is the channel variance.

Reweighting of feature maps: Apply the attention weights to reweight the input feature maps to obtain the output feature maps.

$$X_{\text{att}} = X \times \sigma(E_{\text{inv}}) \tag{12}$$

Thereinto, $\sigma$ is the sigmoid function

**Loss function**

The loss function in this paper consists of two types of loss functions: Dice Loss and Binary Cross-Entropy Loss.

Dice Loss is a loss function based on the Dice Similarity Coefficient (DSC). DSC measures the similarity between two sets and is commonly used in image segmentation to evaluate the degree of overlap between the predicted segmentation result and the ground truth. The formula is as follows:

$$DSC = \frac{2 \cdot |A \cap B|}{|A| + |B|} = \frac{2\sum_i (p_i \cdot g_i)}{\sum_i p_i + \sum_i g_i} \tag{13}$$

Where A and B are two sets, in image segmentation, A is the predicted segmentation result, and B is the ground truth. They correspond to the predicted and true values of the first pixel, respectively.

DICE losses are defined as:

$$\text{DiceLoss} = 1 - DSC = 1 - \frac{2\sum_i (p_i \cdot g_i)}{\sum_i p_i + \sum_i g_i} \tag{14}$$

Binary Cross-Entropy (BCE) Loss quantifies the difference between two probability distributions, and is particularly suitable for binary classification problems. In the image segmentation task, BCE Loss evaluates the difference between the predicted and true values of each pixel. The formula is as follows:

$$\text{BCELoss} = -\frac{1}{N} \sum_{i=1}^{N} [y_i \log(p_i) + (1 - y_i) \log(1 - p_i)] \tag{15}$$

Thereinto, N is the total number of samples, $y_i$ is the real label of the ith sample, $p_i$ is the predicted probability of the first sample.

The loss function used in this paper combines Dice Loss and BCE Loss to optimize both simultaneously and balance their impact on the final segmentation result. The formula for the combined loss function is as follows:

$$\text{Loss} = \alpha \cdot \text{DiceLoss} + (1 - \alpha) \cdot \text{BCELoss} \tag{16}$$

Thereinto, $\alpha$ is the weight factor, Control the contribution of Dice Loss and BCE Loss to the final loss.

## Experiments and analysis

This section describes the dataset used in the experiment. Then, To verify the effectiveness of the module or component, three sets of ablation experiments were were conducted. Finally, the evaluation metrics used are briefly described. All experiments were run in a Pytorch environment on an NVIDIA GeForce RTX 4090TiGPU. The learning rate is set to 1e-4, the batch_size is 2, and the epoch is 50.

## Data set

The dataset used in this paper, EchoNet-Dynamic, comes from the Echocardiography Laboratory and the Center for Artificial Intelligence in Medical Imaging (AIMI) at Stanford University. It is a standard fully resting echocardiogram consisting of 50–100 video and still images that visualize the heart from different angles, positions, and image acquisition techniques (2D images, tissue Doppler images, color Doppler images, etc.). In this dataset, a vertex 4-chamber 2D grayscale video is extracted from each study. Each video represents a unique individual. The representative frames of the EchoNet-Dynamic dataset are displayed in five independent videos, totaling 11 frames after removing ECG data, text labels, and ultrasound acquisition information, as shown in Fig 4. The dataset contains 10,036 echocardiogram videos from 10,036 random individuals who underwent echocardiography between 2006 and 2018.

The apical 4-chamber view video was identified by extracting a medical digital imaging and communication (DICOM) file associated with ventricular volume measurements, which were used to calculate ejection fraction in the apical ventricular view. Each study was linked to clinical measurements and calculations obtained by a registered sonographer and validated by a level 3 echocardiographer in a standard clinical workflow. The left ventricular ejection fraction, a core indicator of cardiac function. It is used to diagnose cardiomyopathy, evaluate eligibility for certain chemotherapy treatments, and determine indications for medical devices. Left ventricular ejection fraction is significantly associated with mortality in

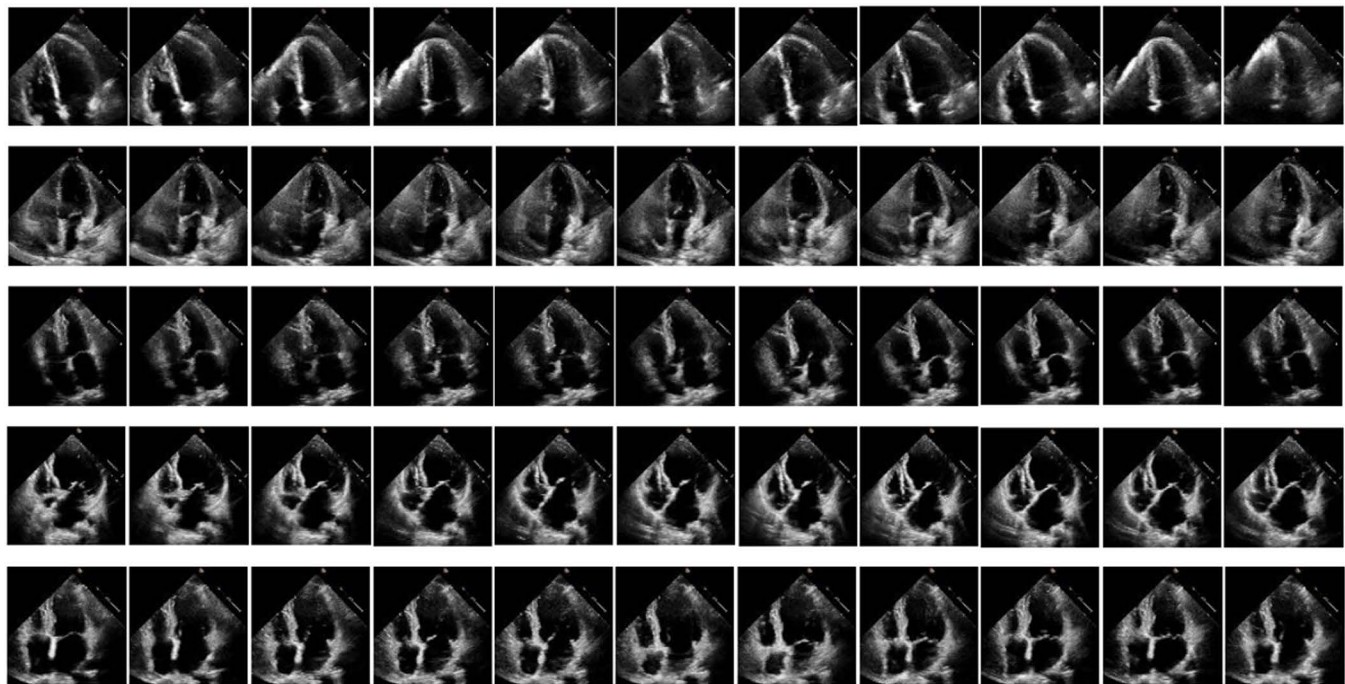

**Fig 4. Representative frames of the dataset.** After removing ECG data, text, and collected information, 11 frames of images were extracted from 5 videos, each representing an independent individual.

many disease states, with a lower ejection fraction correlating with a worse prognosis [32]. The ejection fraction is calculated as (EDV – ESV)/ EDV, as shown in Fig 5. Therefore, this study proposes a DA-UNet++ model to segment more accurate ESV and EDV from video data to calculate left ventricular ejection fraction, providing clinicians with more accurate assistance.

## Ablation experiments

To evaluate the method in this paper and verify its performance under different settings, a series of ablation experiments were conducted. These experiments include variations in attention mechanisms, different scales of tailoring optimization, different loss functions, and the necessity of each module in the overall framework.

### Different kinds of attention mechanisms

This study evaluates the influence of SCBAM components on the model by testing how the model can focus more on important features and locations, thereby improving its performance in image segmentation and detection tasks. The effects of different attention mechanisms on the model are also tested. In this study, the segmentation results of SE, CBAM, ECA, and SCBAM are shown in Table 1. The results show that SCBAM has the best segmentation performance, as it effectively focuses on important features and locations, which leads to superior results.

### Cropping optimization at different scales

To evaluate the impact of different scales of UNet++ on the model, this study tests how reducing the network scale can decrease computing and storage overhead while maintaining accuracy, thereby improving performance in image segmentation and detection tasks. As shown in Table 2, L3 pruning significantly outperforms the unpruned UNet++ (UNet-l4) and the network scale is significantly smaller than that of the traditional UNet++ network, enabling accurate segmentation of video data based on L3 pruning.

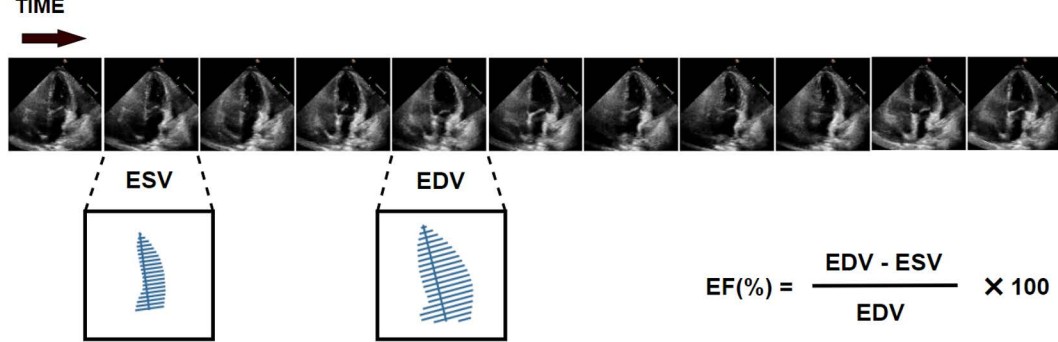

**Fig 5. Calculation of ejection fraction EF.** The calculation formula for ejection fraction is (EDV – ESV)/ EDV.

**Table 1. Analysis of the segmentation performance of different attention mechanisms (%). By comparing the segmentation performance of different attention mechanisms, it is highlighted that SCBAM exhibits better segmentation performance.**

| Attention Mechanism | DSC(%) | IoU(%) |
|---|---|---|
| SE | 0.9293 | 0.4646 |
| CBAM | 0.9294 | 0.4647 |
| ECA | 0.9291 | 0.4645 |
| SCBAM | 0.9316 | 0.4655 |

**Table 2. Comparison of DSC with different cutting scales (%). Evaluating the impact of different scales of UNet++ on the model can reduce the network scale to lower computational and storage overhead while ensuring the accuracy of the model.**

| Model | DSC(%) | IoU(%) |
|---|---|---|
| UNet-l1 | 0.9254 | 0.4627 |
| UNet-l2 | 0.9290 | 0.4645 |
| UNet-l3 | 0.9316 | 0.4655 |
| UNet-l4 | 0.9291 | 0.4645 |

### Different loss functions

To establish a reasonable loss function calculation method, grid search was used to perform ablation experiments on the loss function parameters, as shown in Table 3. The results show that the best performance is achieved by combining Dice Loss and BCE Loss, so this combination is used as the loss function in this study. It is verified that the effect is optimal when the combination of Dice Loss and BCE Loss is set to 0.8, as shown in equation (13).

### The role of each module in the overall framework

Based on UNet++, SCBAM modules, cropping optimization, and NestedUNet nesting modules were added to evaluate the necessity of each component, as shown in Table 4. With the addition of each module, the segmentation results improve, and the complete method achieved the best DSC and IoU results, demonstrating the effectiveness of each module in enhancing segmentation performance.

### Evaluation indicators

The evaluation indicators used in this study are DSC (Dice Similarity Coefficient) and IoU (Intersection over Union), both commonly used in image segmentation tasks as they effectively and intuitively measure segmentation quality. Each indicator has its own unique advantages and applicability in evaluating segmentation results.

**Table 3. Comparison of different loss functions (%). Verify the optimal combination weight of Dice Loss and BCE Loss.**

| Valid values | DSC(%) | IoU(%) |
|---|---|---|
| 0 | 0.9286 | 0.4643 |
| 0.2 | 0.9290 | 0.4645 |
| 0.4 | 0.9288 | 0.4644 |
| 0.6 | 0.9296 | 0.4648 |
| 0.8 | 0.9316 | 0.4655 |
| 1 | 0.9295 | 0.4647 |

**Table 4. The role of each module for the overall framework (%). To evaluate the necessity of each module, the segmentation results improved with the addition of each module, and the complete method achieved the best results.**

| Model | DSC(%) | IoU(%) |
|---|---|---|
| UNet++ | 0.9168 | 0.4584 |
| U+SCBAM | 0.9212 | 0.4606 |
| U+SCBAM+L3 | 0.9266 | 0.4637 |
| U+SCBAM+L3+NEST | 0.9316 | 0.4655 |

DSC is more concerned with the balance of overlapping areas, and its formula is:

$$DSC = \frac{2 \times |A \cap B|}{|A| + |B|}$$

(17)

Thereinto, $A$ is the set of the predicted segmentation results. $B$ is a collection of the true split result. $|A \cap B|$ Indicates the number of elements in the intersection of the predicted result and the true result.

The IoU more rigorously evaluates the degree of matching between the predicted and real regions, and its formula is:

$$IoU = \frac{|A \cap B|}{|A \cup B|}$$

(18)

Thereinto, $|A \cup B|$ indicates the number of elements in the union of the predicted and true results.

## Experimental results

To evaluate the DA-UNet++ architecture, its performance was compared with that of seven other models, performing both quantitative and qualitative analyses of the results.

## Quantitative results

To evaluate the segmentation performance of DA-UNet++ on the EchoNet-Dynamic dataset, comparative experiments were conducted using Deeplabv3, Deeplabv3+, UNet++, UNet-l2, ResNet, FCN, and PSPNet. In the experiment, Dice similarity coefficient and IoU were used as evaluation metrics to measure segmentation accuracy, and the number of floating-point operations (FLOPs) and parameter count (Params) were used to assess computational complexity. The experimental results of DA-UNet++ on the EchoNet-Dynamic dataset are shown in Table 5.

As shown in Table 5, the DA-UNet++ model achieved the best performance in both the Dice Similarity Coefficient (DSC) and Intersection over Union (IoU) on the EchoNet-Dynamic dataset. Specifically, the DSC reached 0.9316 (95% CI: 0.9303–0.9331), and the IoU was 0.4655 (95% CI: 0.4645–0.4664). Although the improvement in DSC over the original Deeplab model was 1.05%, further analysis using confidence intervals demonstrated the statistical significance and robustness of these results. In addition, the DA-UNet++ model's DSC was 1.03% higher than that of the original Deeplab model, while its parameter count was only 15% of Deeplab's, indicating significant efficiency gains.

Comparative analysis of inference speed and computational efficiency was also conducted. On an NVIDIA RTX 4090 GPU, the original Deeplab model achieved an inference speed of 6.48 images per second, with a computational cost of

Table 5. Model comparison. A comparison of DA-UNet++ and other models in terms of parameter quantity and segmentation performance was made, where the abscissa represents the model's parameter quantity, and the ordinate represents the model's Dice similarity coefficient on the EchoNet-Dynamic dataset.

| Model | DSC(%) | IoU(%) | FLOPs | Params |
|---|---|---|---|---|
| Deeplabv3 | 0.9213 | 0.4507 | 41.07 G | 39.63 M |
| Deeplabv3+ | 0.7879 | 0.3940 | 18.61 G | 41.81 M |
| UNet++ | 0.9262 | 0.4613 | 34.76 G | 9.16 M |
| UNet-l2 | 0.9220 | 0.4602 | 10.52 G | **0.52 M** |
| ResNet34 | 0.9215 | 0.4607 | **7.69 G** | 24.21 M |
| FCN | 0.9202 | 0.4601 | 34.75 **G** | 32.95 M |
| PSPNet | 0.8480 | 0.4240 | 13.35 G | 46.58 M |
| DA-UNet++ | **0.9316** | **0.4655** | 25.74 G | 5.96 M |

34.76 GFLOPs and 9.16 million parameters. In contrast, the improved DA-UNet++ model increased the inference speed to 7.95 images per second, reduced computational cost to 25.74 GFLOPs, and decreased the parameter count to 5.96 million. These results demonstrate that the proposed model not only substantially reduces computational complexity but also significantly improves inference speed, making it better suited for practical deployment.

To visually illustrate the advantages of DA-UNet++ over other models, Fig 6 compares the models in terms of parameter quantity (x-axis) and segmentation performance (DSC, y-axis) on the EchoNet-Dynamic dataset. As shown, DA-UNet++ achieves the best segmentation performance while consuming fewer computing resources than competing models. Notably, although the UNet-l2 model has the fewest parameters, its DSC is only 0.07% higher than that of the traditional model, further highlighting the superior balance of efficiency and performance achieved by DA-UNet++.

**Qualitative results**

In order to demonstrate the effect of this method on the segmentation of the left ventricle with variable shape and size, this study used six different models to segment the images, and selected the images of two individuals for display. This is shown in Figs 7 and 8, Figs 7b and 8b are the EDV and ESV of the heart structure outline annotated by the original

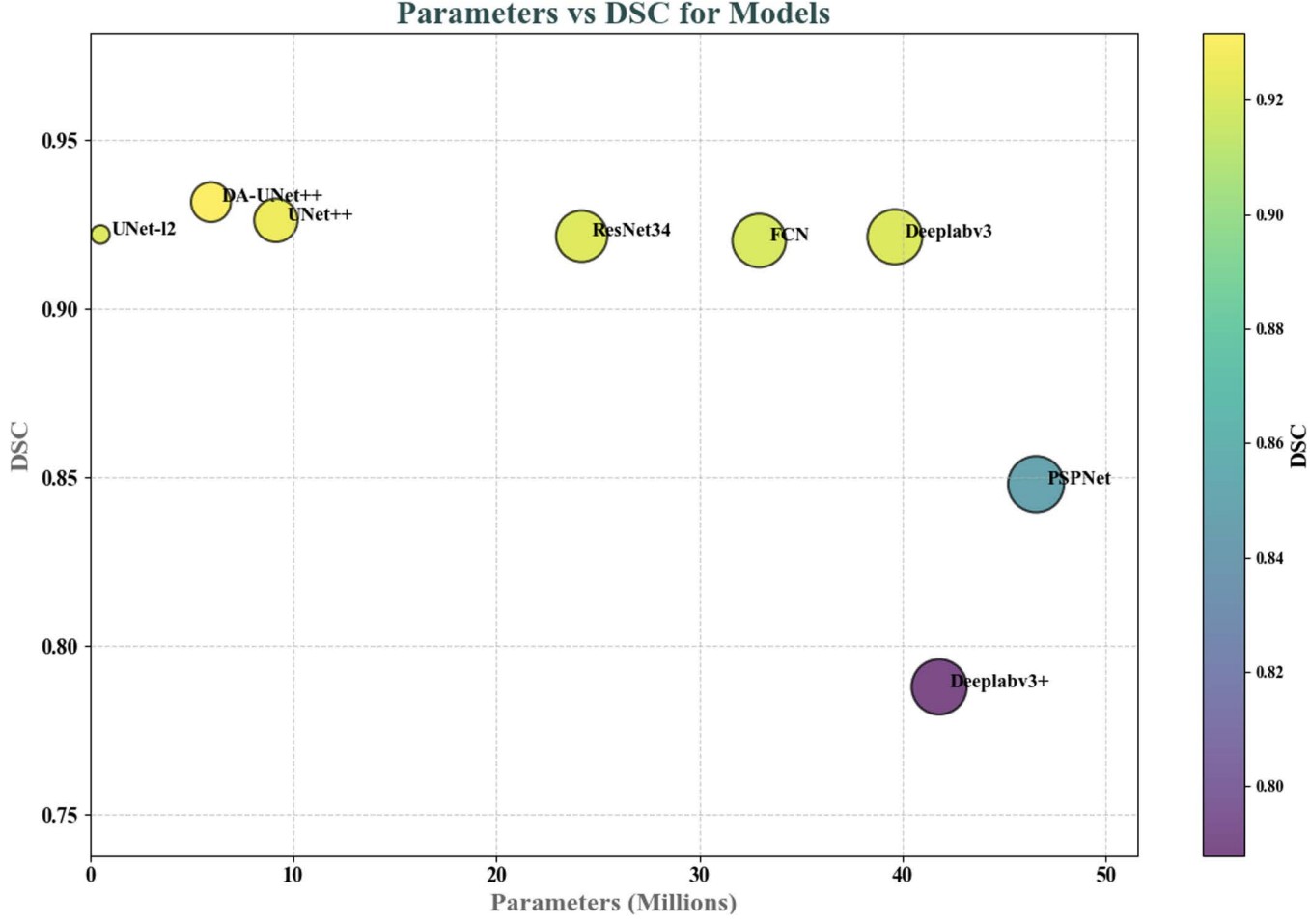

**Fig 6. Compared with other model.** The comparison between DA-UNet+ and other models in terms of parameter count and segmentation performance.

model, respectively, and Figs 7g and 8g are the EDV and ESV of the heart outline annotated using the method in this paper, respectively. Compared with the previous method, the DSC of the proposed method reached 93.16%, indicating that the proposed method was more accurate in cardiac segmentation. It can be seen from the visualization results that the proposed method is more complete for the segmentation of the left ventricle, which can effectively solve the problem of large changes in the scale of the left ventricle and improve the accuracy of left ventricular segmentation.

## Conclusion

In order to solve the problems of variable ventricular shape and size and blurred tissue boundaries, a fusion and improved DA-UNet++ model for left ventricular segmentation was proposed. The algorithm optimizes the UNet module, integrates the NestedUNet network, extracts rich global and local features, and reduces the computational and storage overhead. In addition, the model integrates the SCBAM attention mechanism, which recalibrates the features while increasing the information extraction ability and encoding multi-scale feature information, allowing the network to pay more attention to

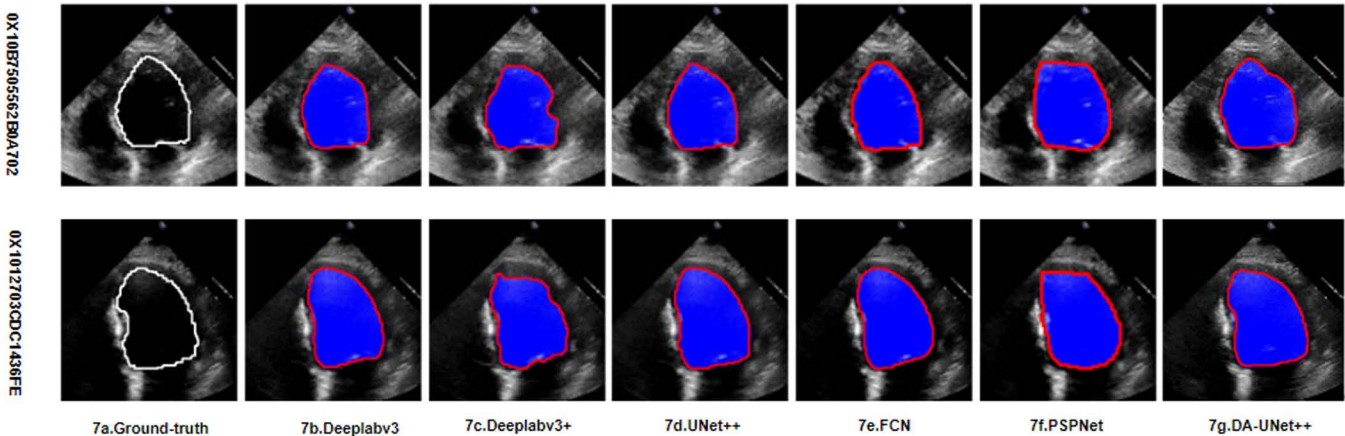

**Fig 7. EDV of the left ventricle, The white line represents the ground truth, and the red line represents the prediction of the current model (i.e., the model labeled in the figure).**

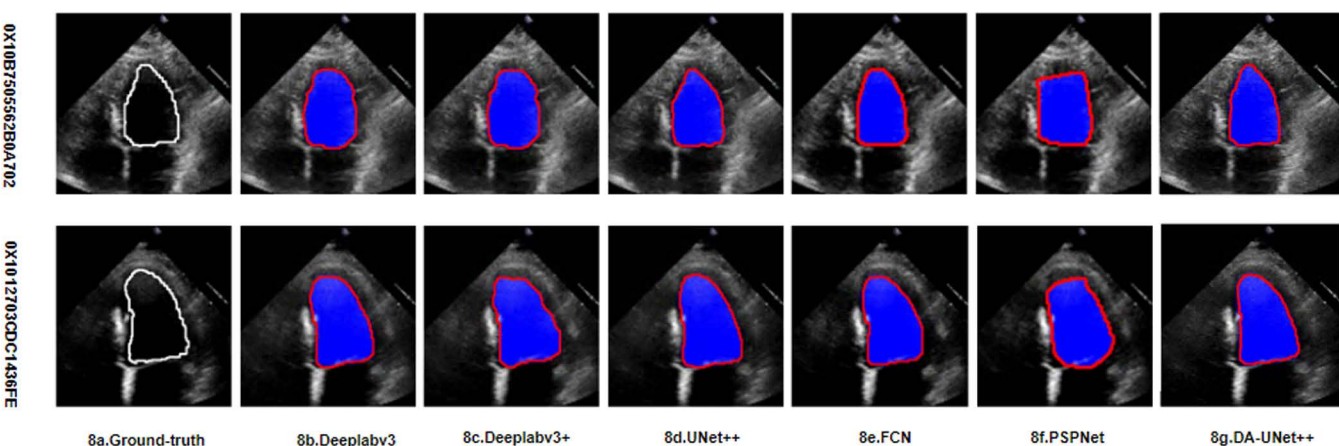

**Fig 8. ESV of the left ventricle, The white line represents the ground truth, and the red line represents the prediction of the current model (i.e., the model labeled in the figure).**

useful features, thereby enhancing the feature extraction ability of ventricular video data and improving the performance of ventricular segmentation. The segmentation performance of the proposed method was verified and evaluated on the EchoNet-Dynamic dataset, and the results showed that compared with the original model DeepLabv3 for medical image segmentation, the DSC was improved by 1.03%, and the model size was only 15% of the original model. This paper aims to solve the problem of limited practical medical auxiliary applications due to the high demand for computing resources and the large scale of the model in clinical applications, which has certain clinical application value.

While the EchoNet-Dynamic dataset provides a robust foundation for model training and evaluation, generalization to other medical datasets remains a key concern. The Pruned Nested UNet with SCBAM demonstrates versatility, making it suitable for various medical image segmentation tasks requiring fine feature extraction and attention mechanisms. However, challenges remain when applying the model to new datasets. Many medical datasets, especially in specialized fields like echocardiography, are limited in size, resolution, or annotation quality. Additionally, models trained on one type of imaging, such as cardiac ultrasound, may not transfer well to others like X-rays or MRIs due to differences in contrast, noise, and resolution. Furthermore, while attention-based models excel in performance, their interpretability still needs improvement, a key requirement for clinical applications.

Future work will focus on the following aspects to further enhance the clinical value and applicability of the model: (1) Evaluating the model's generalization performance on other imaging modalities such as MRI and CT to verify its cross-modality adaptability; (2) Exploring deployment and optimization on edge devices (e.g., portable ultrasound equipment) to achieve real-time and efficient segmentation; (3) Improving the interpretability of the model to enhance its credibility and applicability in clinical practice. Through these research directions, the DA-UNet++ model is expected to expand its application scope and practical value in a variety of medical imaging scenarios.

## Supporting information

**S1 File. The main code of this article.**
(ZIP)

**S2 File. Dataset.**
(ZIP)

## Author contributions

**Software:** Minghui Geng.

**Visualization:** Shuai Zheng.

**Writing – original draft:** Kerang Cao, Miao Zhao.

**Writing – review & editing:** Kerang Cao, MIAO Zhao, Hoekyung Jung.

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
