## [Decision Letter · Decision Letter 0]

Dear Dr. jung,

Thank you for submitting your manuscript to PLOS ONE. After careful consideration, we feel that it has merit but does not fully meet PLOS ONE’s publication criteria as it currently stands. Therefore, we invite you to submit a revised version of the manuscript that addresses the points raised during the review process.

We look forward to receiving your revised manuscript.

Kind regards,

Mohamed Yacin Sikkandar

Academic Editor

PLOS ONE

Journal Requirements:

2. We suggest you thoroughly copyedit your manuscript for language usage, spelling, and grammar. If you do not know anyone who can help you do this, you may wish to consider employing a professional scientific editing service. The American Journal Experts (AJE) (https://www.aje.com/) is one such service that has extensive experience helping authors meet PLOS guidelines and can provide language editing, translation, manuscript formatting, and figure formatting to ensure your manuscript meets our submission guidelines. Please note that having the manuscript copyedited by AJE or any other editing services does not guarantee selection for peer review or acceptance for publication. Upon resubmission, please provide the following: ● The name of the colleague or the details of the professional service that edited your manuscript ● A copy of your manuscript showing your changes by either highlighting them or using track changes (uploaded as a *supporting information* file) ● A clean copy of the edited manuscript (uploaded as the new *manuscript* file)

4. Thank you for stating the following financial disclosure: “This research was supported by the MSIT(Ministry of Science and ICT), Korea, under the Innovative Human Resource Development for Local Intellectualization support pro-gram (IITP-2024-RS-2022-00156334) supervised by the IITP(Institute for Information & communi-cations Technology Planning & Evaluation).This research was support by “Regional Innovation Strategy(RIS)”, through the Nation Research Foundation of Korea(NRF) funded by the Ministry of Education(MOE2021RIS-004);Basic scientific research general project of Liaoning Provincial Department of Education in 2023(JYTMS20231518)”

5. Please upload a copy of Figures 1-8, to which you refer in your text on page 17-31. If the figure is no longer to be included as part of the submission please remove all reference to it within the text.

**Dear Authors,**
**Please address the reviewers comments.**

Reviewers' comments:

Reviewer's Responses to Questions

**Comments to the Author**

1. Is the manuscript technically sound, and do the data support the conclusions?

Reviewer #1: Yes

Reviewer #2: Partly

2. Has the statistical analysis been performed appropriately and rigorously?

Reviewer #1: Yes

Reviewer #2: Yes

3. Have the authors made all data underlying the findings in their manuscript fully available?

Reviewer #1: Yes

Reviewer #2: Yes

4. Is the manuscript presented in an intelligible fashion and written in standard English?

Reviewer #1: Yes

Reviewer #2: Yes

Reviewer #1: 1.Simplify overly complex sentences. For example, instead of "These need to be refined for clarity and improved flow," consider "These should be clearer and easier to read." This avoids unnecessary complication while maintaining meaning. Correct grammatical errors to ensure smooth, intuitive reading. Alignment is not good. Avoid the usage of We / I in the article. While using the Acronyms first time, expand them. For example CBAM in abstract.

2. For mechanisms like CBAM and SimAM, providing more context is key. Explain them in simpler terms, offering real-world analogies if possible. This will help a wider audience grasp the technical aspects without needing a specialized background.

3.The addition of attention mechanisms may slow things down, especially in real-time applications. Include a discussion on how to tweak or optimize the model so it remains effective for real-time use without losing too much of its complexity. This section should speak directly to practical challenges and solutions.

4.The EchoNet-Dynamic dataset is central to the paper, but expanding the discussion to how the model might work with other datasets—specifically in medical contexts—will make the findings more impactful. A brief note on cross-validation or potential limitations would be helpful here

5. What are all the features extracted? How many are used for segmentation? What is the reason behind the selection of features?

6.While you’ve included numbers and metrics, visuals can tell a different part of the story. Show side-by-side comparisons of segmentation results from different models. These visual cues will help readers quickly see the strengths of your proposed method.

Reviewer #2: Concept, Figures and Tables are given and explained clearly.

Many grammatical mistakes are there

The author have not shown any interest in formatting the document

Fig. 6 have to be regenerated - Comparisons of the other model

inside fig 7 the sub division of figures mentioned as 6a.,6b. same as Fig8. Clarity Needed

**Do you want your identity to be public for this peer review?** For information about this choice, including consent withdrawal, please see our Privacy Policy

Reviewer #1: **Yes: ** M.C.JOBIN CHRIST

Reviewer #2: No

---

## [Author Response · Author response to Decision Letter 1]

10 Dec 2024

Dear reviewer,

We sincerely thank the editor and all reviewers for their valuable feedback that we have used to improve the quality of our manuscript. The reviewer comments are laid out below in italicized font and specific concern shave been numbered. Our response is given in normal font and changes/additions to the manuscript are given in Highlight Text.

Reviewer 1

1.Simplify overly complex sentences. For example, instead of "These need to be refined for clarity and improved flow," consider "These should be clearer and easier to read." This avoids unnecessary complication while maintaining meaning. Correct grammatical errors to ensure smooth, intuitive reading. Alignment is not good. Avoid the usage of We / I in the article. While using the Acronyms first time, expand them. For example CBAM in abstract.

Response: Thank you for your thorough review of our paper and for providing many valuable comments. We highly value your feedback and have diligently addressed the issues you raised during the revision process. In response to the specific points you identified, we have taken the following measures:

1. Simplified Complex Sentences: We have simplified the complex sentences in the manuscript to ensure that the expression is clearer and more concise. We have avoided lengthy and hard-to-understand sentence structures, ensuring that each sentence is succinct and can smoothly convey the research content.

2. Corrected Grammatical Errors: We have conducted a comprehensive grammar check of the entire manuscript to ensure that no grammatical errors were overlooked. To improve the language quality, we used professional grammar checking tools and also performed manual proofreading to ensure that every sentence adheres to grammatical rules.

3. Improved Readability and Intuitiveness: In the process of simplifying sentences, we paid special attention to the logical structure of the manuscript and the fluency of the language to ensure that the overall expression is natural and smooth, avoiding confusion for readers. Additionally, we ensured smoother transitions between different sections to enhance the readability of the paper.

4. Adjusted Alignment Issues: We have carefully reviewed the formatting of the entire manuscript to ensure that paragraphs, titles, tables, and figures are aligned according to the journal's requirements. We have addressed any non-standard alignment issues to improve the overall layout of the paper.

5. Avoided Using "We/I": We have noted that avoiding the use of the first person is a common norm in academic writing. Therefore, we adjusted the relevant sections of the manuscript to use more objective expressions, avoiding the use of "We" or "I," and ensuring a more neutral and formal tone.

6. Expanded Explanations of Technical Terms: We have provided detailed explanations for technical terms the first time they appear in the manuscript, such as CBAM, UNet, SimAM, etc., to ensure that readers have a clearer understanding of the content.

We have comprehensively revised the manuscript based on your feedback and ensured that these issues have been properly addressed. Once again, thank you for your careful review of our paper and your valuable suggestions. We have completed these revisions and submitted the revised version, looking forward to your further review.

If you have any other suggestions or requests, we are very happy to continue making improvements.

2. For mechanisms like CBAM and SimAM, providing more context is key. Explain them in simpler terms, offering real-world analogies if possible. This will help a wider audience grasp the technical aspects without needing a specialized background.

Response: Thank you for your valuable suggestions regarding our paper. We fully understand the importance of providing more context and using simpler terminology to clearly explain our research content. To enhance the readability and comprehensibility of the paper, we have implemented the following improvements:

1. Using Simpler Terminology: We have reviewed the manuscript for any overly technical or obscure terms and have replaced them with simpler, more easily understandable language to ensure that the paper is accessible to a broad readership. For instance, we have explained some highly specialized terms to prevent any confusion among readers.

2. Providing Real-World Analogies: We have attempted to clarify some abstract or complex concepts by using real-world analogies. By drawing parallels between our theoretical or experimental results and everyday experiences, we aim to make the research content more intuitive and easier to grasp. For example, when explaining CBAM, we have likened it to observing a busy street filled with people, vehicles, and buildings. If you are looking for a friend, your brain naturally focuses on the crowd, ignoring irrelevant elements like parked cars or buildings.

3. Simplifying Expression: In addition to supplementing terminology and context, we have simplified certain sentence structures within the manuscript to make the expression more direct and clear. This involves avoiding overly complex sentence constructions to ensure the paper flows more smoothly and is easier to read.

We believe these improvements have contributed to enhancing the paper's readability and comprehensibility, ensuring that readers can effortlessly understand the research content and conclusions with minimal confusion. Once again, thank you for your invaluable suggestions. Your feedback has been instrumental in helping us improve the quality of our paper. We have promptly completed these modifications and submitted the revised version, looking forward to your further review.

If you have any additional suggestions or requests, please feel free to let us know, and we will continue to make improvements.

P8

Convolutional Block Attention Module (CBAM) is an attention mechanism for convolutional neural networks, which combines channel attention and spatial attention to enhance the representation ability of feature maps. CBAM improves the performance of the model by adaptively weighting the input feature map to highlight the important features and suppress irrelevant ones. CBAM consists of two sub-modules: the Channel Attention Module (CAM) and the Spatial Attention Module (SAM), as shown in Fig 3. CBAM is similar to how our brains focus attention on important parts when observing something. Imagine you are looking at a busy street filled with people, vehicles, and buildings. If you are searching for a friend, your brain naturally concentrates on the crowd, ignoring irrelevant elements like parked cars or buildings. This is exactly what CBAM does in neural networks—it helps the model focus on "important" features, such as distinguishing people from vehicles in an image.

P10

The SimAM (Simple, Parameter-Free Attention Module) [9] design concept is derived from neuroscience theory, aiming to enhance which aims to enhance the representation ability of convolutional neural networks while remaining lightweight without adding additional parameters. SimAM solves two major problems with existing attention modules: attention weights are limited to either the channel or spatial dimensions, and there is a lack of flexibility, requiring additional parameters that increase model complexity. Imagine being in a classroom where a teacher is explaining a topic. Some students (features) are more focused because they have studied more and understood the material more deeply. SimAM is like a teacher who can recognize which students are more attentive and concentrate more attention on those students, thereby enhancing the overall learning effectiveness. It gives more attention to students who have a deeper understanding (important features) while ignoring those who are less engaged (irrelevant features).

3.The addition of attention mechanisms may slow things down, especially in real-time applications. Include a discussion on how to tweak or optimize the model so it remains effective for real-time use without losing too much of its complexity. This section should speak directly to practical challenges and solutions.

Response: Thank you for your valuable suggestions on our paper. Based on your feedback, we have refined the paper to focus more on the practical challenges and our proposed solutions, thereby enhancing its relevance and practicality.

To avoid a significant reduction in inference speed when incorporating attention mechanisms (SCBAM), we have adopted the following optimization strategies:

1. In the PrunedNestedUNet structure, we utilized a nested U-Net (DA-UNet++), a variant of U-Net designed to extract features at multiple scales through several convolutional blocks and upsampling operations. By reducing the number of convolutional layers or channels within the layers, we further minimized computational overhead. Specifically, we pruned redundant layers or channels in the U-Net, removing unnecessary convolutional kernels and neurons. This approach reduced the model size and computational complexity, thereby improving inference speed.

2. When applying attention mechanisms, we reduced computational costs by adjusting the kernel size of the attention module. For example, in the spatial attention module of CBAM (Convolutional Block Attention Module), we reduced the kernel size of the convolution operation, alleviating computational burden while still retaining a reasonable level of attention effectiveness.

We believe these improvements have significantly enhanced the readability and comprehensibility of the paper, ensuring that readers can intuitively understand the core content and solutions of our research. Once again, we sincerely appreciate your valuable feedback, which has been instrumental in improving the quality of our paper. The revised version has been submitted promptly, and we look forward to your further review.

If you have any additional suggestions or requirements, please do not hesitate to inform us. We will continue to make improvements.

P8

By combining these attention mechanisms, the model can more effectively focus on key regions and important channels in the image, thereby improving segmentation accuracy. To prevent the SCBAM attention mechanism from significantly reducing the model's inference speed, this study employs the following optimization methods:

Pruned Nested U-Net Structure: By reducing the number of convolutional layers and pruning redundant convolutional kernels and neurons, the computational overhead is decreased, the model size is reduced, and inference speed is enhanced.

Adjusting the Kernel Size of the Attention Mechanism: When using the spatial attention module of CBAM, smaller convolutional kernels are employed to reduce the computational burden while maintaining effective attention mechanisms.

4.The EchoNet-Dynamic dataset is central to the paper, but expanding the discussion to how the model might work with other datasets—specifically in medical contexts—will make the findings more impactful. A brief note on cross-validation or potential limitations would be helpful here

Response: Thank you for your valuable suggestions on our paper. Based on your feedback, we have further expanded the discussion on the model’s applications in the medical field, particularly focusing on how it can collaborate with other datasets to enhance the impact of the research results.

First, although the EchoNet-Dynamic dataset has provided a solid foundation for training and evaluating the model, we recognize that the model’s performance on other medical datasets is equally important, especially in the medical field where the model’s generalization ability is crucial. The pruned nested U-Net architecture used in this study, combined with SCBAM (Convolutional Block Attention Module), has demonstrated good versatility, making it suitable for other medical image segmentation tasks that require fine-grained feature extraction and attention mechanisms.

However, despite the excellent performance of the model on the EchoNet-Dynamic dataset, we also realize that there may be potential limitations when applying the model to new datasets or other medical imaging tasks. For example, many medical datasets, especially those from specialized fields such as echocardiography, may be limited in terms of scale and quality, with lower resolution or incomplete annotations, which could affect the model’s performance. Furthermore, models trained specifically for one type of medical imaging data may not transfer directly to other domains. For instance, X-ray or MRI images differ significantly from cardiac ultrasound images in contrast, noise, and resolution, which may present challenges when trying to transfer the model.

Additionally, while attention-based models have performed well in various tasks, their interpretability remains a critical issue in the medical field. Clinical decision-making often relies on a high level of trust and transparency in the model’s outputs, making the improvement of model interpretability essential for its broad application.

To address these challenges, we will continue to conduct in-depth research to explore solutions and improve the model’s applicability and reliability across different medical applications. We appreciate your invaluable feedback and look forward to your further review of the revised version.

If you have any additional questions or requests, please feel free to let us know, and we will continue to make improvements.

P17

While the EchoNet-Dynamic dataset provides a robust foundation for model training and evaluation, generalization to other medical datasets remains a key concern The Pruned Nested U-Net with SCBAM demonstrates versatility, making it suitable for various medical image segmentation tasks requiring fine feature extraction and attention mechanisms. However, challenges remain when applying the model to new datasets. Many medical datasets, especially in specialized fields like echocardiography, are limited in size, resolution, or annotation quality. Additionally, models trained on one type of imaging, such as cardiac ultrasound, may not transfer well to others like X-rays or MRIs due to differences in contrast, noise, and resolution. Furthermore, while attention-based models excel in performance, their interpretability still needs improvement, a key requirement for clinical applications. Future work will focus on addressing these limitations to enhance the model's generalizability and reliability in diverse medical contexts.

5.What are all the features extracted? How many are used for segmentation? What is the reason behind the selection of features?

Response: Thank you for your valuable suggestions. Based on your feedback, we will explicitly explain the reasons for choosing the Nested U-Net architecture, CBAM (Convolutional Block Attention Module), and SimAM (Similarity Attention Module) in our paper, and provide a detailed discussion on their application value in our research.

Feature extraction is primarily performed through the Nested U-Net architecture, combined with the attention mechanisms SCBAM (CBAM and SimAM). These features are used to guide the segmentation process with the goal of helping the model focus on the most important regions of the image (the regions that need segmentation). Below is an overview of feature extraction and its use in segmentation:

1. Feature Extraction in the Model Architecture

The Nested U-Net architecture extracts hierarchical features through convolutional layers at different depths, capturing both local and global spatial information in the image.

CBAM extracts channel attention features and spatial attention features. Channel attention focuses on the most important channels in the feature map, enhancing useful feature channels. Spatial attention focuses on specific regions in the image (spatial dimension), enhancing spatial features.

SimAM (Self-Information Adaptive Module) extracts adaptive features by focusing on the variance in the feature map. It computes an "energy function" to independently adjust the attention of each feature map.

These components collectively ensure that the model can focus on important channels and specific spatial regions in the image, thus improving segmentation performance.

2. Features Used for Segme

---

## [Decision Letter · Decision Letter 1]

Dear Dr. jung,

Thank you for submitting your manuscript to PLOS ONE. After careful consideration, we feel that it has merit but does not fully meet PLOS ONE’s publication criteria as it currently stands. Therefore, we invite you to submit a revised version of the manuscript that addresses the points raised during the review process.

We look forward to receiving your revised manuscript.

Kind regards,

Mohamed Yacin Sikkandar

Academic Editor

PLOS ONE

Journal Requirements:

Reviewers' comments:

Reviewer's Responses to Questions

**Comments to the Author**

Reviewer #1: All comments have been addressed

Reviewer #3: All comments have been addressed

2. Is the manuscript technically sound, and do the data support the conclusions?

Reviewer #1: Yes

Reviewer #3: Yes

3. Has the statistical analysis been performed appropriately and rigorously?

Reviewer #1: Yes

Reviewer #3: Yes

4. Have the authors made all data underlying the findings in their manuscript fully available?

Reviewer #1: Yes

Reviewer #3: Yes

5. Is the manuscript presented in an intelligible fashion and written in standard English?

Reviewer #1: No

Reviewer #3: Yes

Reviewer #1: 1. Words are frequently hyphenated at the end of lines, with the second part appearing at the beginning of the next line. This occurs throughout the article.

2. Table 4 uses a different font.

3. The legend of Figure 6 uses a different font.

4. The article lacks proper alignment.

5. The right side of Figure 1 (the yellow block diagram) is not clear.

6. The reference section does not provide complete author lists. "Et al." is used in many places, the full list should be provided.

Reviewer #3: The reported 1.05% improvement in Dice is modest—statistical significance tests (e.g., p-values or confidence intervals) would strengthen this.

Visual comparisons (e.g., segmentation overlays) were promised but could be more prominently and consistently used.

Inference speed or runtime performance is not clearly benchmarked—important for real-time clinical utility claims

Overuse of passive voice and vague terms (e.g., “it is proposed that…”).

Some paragraphs are overly long and would benefit from clearer topic sentences and transitions.

Inconsistent spacing, alignment, and formatting across sections.

Some promises made in the response (e.g., simplified figures, added analogies) should be double-checked in the revised manuscript to ensure they were implemented.

Briefly mention future directions (e.g., generalizability testing on MRI or CT, or deployment in edge devices).

**Do you want your identity to be public for this peer review?** For information about this choice, including consent withdrawal, please see our Privacy Policy

Reviewer #1: **Yes: ** M.C.JOBIN CHRIST

Reviewer #3: No

---

## [Author Response · Author response to Decision Letter 2]

17 May 2025

MIAO ZHAO

Shenyang University of Chemical Technology,Shenyang

Liaoling,110142,

China

Email:a17642058813@gmail.com

May 17, 2025

Dear editor,

On behalf of all coauthors, I am submitting the revised version of our manuscript entitled “Left ventricular segmentation method based on optimized UNet and improved CBAM: ESV and EDV tracking study” for your consideration for publication in PLOS Computational Biology. We have carefully addressed the feedback provided by the reviewers and made revisions to improve the clarity, flow, and technical depth of the manuscript.

Thank you very much for your valuable feedback and constructive suggestions. We have carefully addressed all comments to improve the quality and clarity of our manuscript. Below is a summary of the main revisions based on the reviewers’ feedback:

Formatting and Consistency: All issues related to hyphenation, font usage (including Table 4 and figure legends), alignment, and overall formatting have been resolved to ensure a unified and professional appearance throughout the manuscript.

Figures and Visuals: Figures 1, 6, 7, and 8 have been revised for greater clarity and consistency. We improved resolution, updated color schemes, and enhanced legends to make visual comparisons clearer and easier to interpret.

References: All references have been carefully checked and updated to provide complete author lists, replacing "et al." where required, in accordance with journal requirements. We confirm that no references were changed, removed, or replaced; only the author lists were completed to ensure full and accurate attribution.

Technical Improvements: We have incorporated statistical significance tests and confidence intervals for key results (such as the Dice coefficient), and expanded our discussion of model performance metrics, including inference speed and computational complexity.

Content Clarity: We have shortened overly long paragraphs, improved topic sentences and transitions, and revised passive or vague language for clarity and precision. Real-world analogies for CBAM and SimAM have also been included to facilitate understanding.

Future Directions: The revised manuscript now briefly discusses future research, such as generalizability testing on other datasets (e.g., MRI, CT) and potential deployment on edge devices.

We believe that these revisions have thoroughly addressed all the concerns raised by the reviewers and have significantly improved the quality and clarity of the manuscript. We feel that the revised version is now more refined and suitable for publication in PLOS Computational Biology, and we hope it will attract the interest of a broad audience, including radiologists, clinical researchers, computer scientists, and others working in related fields.

Thank you for considering our revised manuscript. We look forward to your feedback and hope for a positive outcome.

Sincerely yours,

MIAO ZHAO

Shenyang University of Chemical Technology,Shenyang

---

## [Editor Report · Decision Letter 2]

Left ventricular segmentation method based on optimized UNet and improved CBAM: ESV and EDV tracking study

PONE-D-24-45156R2

Dear Dr. jung,

We’re pleased to inform you that your manuscript has been judged scientifically suitable for publication and will be formally accepted for publication once it meets all outstanding technical requirements.

Kind regards,

Mohamed Yacin Sikkandar

Academic Editor

PLOS ONE
---

## [Editor Report · Acceptance letter]

PONE-D-24-45156R2

PLOS ONE

Dear Dr. JUNG,

I'm pleased to inform you that your manuscript has been deemed suitable for publication in PLOS ONE. Congratulations! Your manuscript is now being handed over to our production team.

Kind regards,

on behalf of

Dr. Mohamed Yacin Sikkandar

Academic Editor

PLOS ONE